# Immune Responses to Irradiated Pneumococcal Whole Cell Vaccine

**DOI:** 10.3390/vaccines9040405

**Published:** 2021-04-19

**Authors:** Eunbyeol Ko, Soyoung Jeong, Min Yong Jwa, A Reum Kim, Ye-Eun Ha, Sun Kyung Kim, Sungho Jeong, Ki Bum Ahn, Ho Seong Seo, Cheol-Heui Yun, Seung Hyun Han

**Affiliations:** 1Department of Oral Microbiology and Immunology and DRI, School of Dentistry, Seoul National University, Seoul 08826, Korea; whitebmilk@snu.ac.kr (E.K.); syjeong94@snu.ac.kr (S.J.); myou72j@snu.ac.kr (M.Y.J.); kimareum@snu.ac.kr (AR.K.); pure9502@naver.com (Y.-E.H.); burnedpooh@gmail.com (S.K.K.); ssgd33@snu.ac.kr (S.J.); 2Radiation Research Division, Korea Atomic Energy Research Institute, Jeongeup 56212, Korea; ahnkb@kaeri.re.kr (K.B.A.); hoseongseo@kaeri.re.kr (H.S.S.); 3Department of Radiation Biotechnology and Applied Radioisotope Science, University of Science and Technology, Daejeon 34113, Korea; 4Department of Agricultural Biotechnology and Research Institute of Agriculture and Life Sciences, Seoul National University, Seoul 08826, Korea; cyun@snu.ac.kr

**Keywords:** *Streptococcus pneumoniae*, gamma-irradiation, whole-cell vaccine, intranasal immunization, immune responses

## Abstract

*Streptococcus pneumoniae* (pneumococcus) can cause respiratory and systemic diseases. Recently, γ-irradiation-inactivated, non-encapsulated, intranasal *S. pneumoniae* (r-SP) vaccine has been introduced as a novel serotype-independent and cost-effective vaccine. However, the immunogenic mechanism of r-SP is poorly understood. Here, we comparatively investigated the protective immunity and immunogenicity of r-SP to the heat-(h-SP) or formalin-inactivated vaccine (f-SP) without adjuvants. Mice were intranasally immunized with each vaccine three times and then challenged with a lethal dose of *S. pneumoniae* TIGR4 strain and then subsequently evaluated for their immune responses. Immunization with r-SP elicited modestly higher protection against *S. pneumoniae* than h-SP or f-SP. Immunization with r-SP enhanced pneumococcal-specific IgA in the nasal wash and IgG in bronchoalveolar lavage fluid. Immunization with r-SP enhanced *S. pneumoniae*-specific IgG, IgG1, and IgG2b in the serum. r-SP more potently induced the maturation of dendritic cells in the cervical lymph nodes than h-SP or f-SP. Interestingly, populations of follicular helper T cells and IL-4-producing cells were potently increased in cervical lymph nodes of r-SP-immunized mice. Collectively, r-SP could be an effective intranasal, inactivated whole-cell vaccine in that it elicits *S. pneumoniae*-specific antibody production and follicular helper T cell activation leading to protective immune responses against *S. pneumoniae* infection.

## 1. Introduction

*Streptococcus pneumoniae* (*S. pneumoniae*; pneumococcus) is a facultative anaerobic Gram-positive bacterium that can reside asymptomatically in healthy carriers, normally colonizing in the nasopharynx. However, in susceptible individuals, the asymptomatic colonization may become pathogenic, lead to bacterial dissemination, and cause respiratory and systemic diseases [1]. Of the many diseases caused by *S. pneumoniae*, pneumonia is one of its most life-threatening presentations, especially among the elderly, children, and immunocompromised patients [1,2]. Pneumococcal pneumonia is a major cause of global hospitalization and mortality. It results in approximately 800,000 deaths per year among children [3,4]. A number of pneumococcal virulence factors have been reported, including capsular polysaccharides (C-PS), pneumolysin, lipoproteins, and cell surface anchoring LPXTG motif proteins [5,6,7]. Among them, C-PS is known to interfere with the phagocytic activity of the host immune cells and anti-C-PS antibodies confer protective immunity against pneumococcal infections. Therefore, it has become a major target of the pneumococcal vaccine [8]. More than 98 pneumococcal C-PS types have been reported. Its structural variations account for different immunological serotypes of pneumococcus [9].

Due to the emergence of antibiotic-resistant *S. pneumoniae*, preventing its infection through vaccination has put the pressure on public health sectors to minimize pneumococcal disease [10,11]. Currently, there are two types of pneumococcal vaccines which are the pneumococcal polysaccharide vaccine (PPSV) and pneumococcal conjugate vaccine (PCV). PPSV successfully vaccinates against 23 common serotypes in adults, but the vaccine fails to induce long immunological memory due to its T cell-independent immune responses [12]. On the other hand, PCV contains immunogenic carrier proteins that induce T cell-dependent immune responses, allowing robust immune responses and immunological memory in both children and adults [13,14]. However, some of the critical limitations of PCV, particularly in low-income countries, include its high cost and limited coverage of the pneumococcal serotypes [15]. As a matter of fact, *S. pneumoniae* infections remain the leading cause of child death in developing countries despite the presence of both vaccines [15,16]. Therefore, there is an urgent need for a cost-effective vaccine which elicits serotype-independent immune response.

The currently available vaccines are largely classified into the following four categories based on their formulation: inactivated whole-cell, live-attenuated, subunit, and conjugate vaccines. Among these vaccine types, inactivated whole-cell vaccines are one of the most convenient and cost-effective vaccines against bacterial infections [17]. These vaccines are stable, safe, and can be produced relatively rapidly at low cost [18]. Whole cells are commonly inactivated with heat or formalin treatment. However, both methods may modify or damage key antigens that are needed to evoke protective immune responses, leading to unexpectedly low or reduced immune responses [19,20]. Recently, gamma-irradiation has been considered as an alternative bacterial inactivation method to produce a whole-cell vaccine. This method produces potent intracellular reactive oxygen species (ROS), which can primarily destroy nucleic acids and cause damage to the molecular structure to surface antigens [19,21,22].

*S. pneumoniae* colonizes the mucosal surface of the upper respiratory tract asymptomatically [6]. It later migrates through the mucosal surface to the lungs and invades other organs when the host has become immunocompromised [1]. Despite its need, currently available pneumococcal vaccines cannot elicit a robust mucosal immune response. For this reason, whole-cell inactivated mucosal vaccines are being considered as alternative vaccines for serotype-independent prevention against pneumococcal diseases. Recently, the intranasal administration of gamma-irradiated *S. pneumoniae* has been proposed as a serotype-independent whole-cell vaccine to elicit mucosal and systemic immune responses [23]. Our group also previously reported that gamma-irradiated, non-encapsulated *S. pneumoniae* (r-SP) immunization with cholera toxin (CT) as an adjuvant results in greater antibody production than does heat- or formalin-inactivated *S. pneumoniae* (h-SP or f-SP) immunization with CT in mice [24]. In this study, we investigated the capacity of the r-SP vaccine to induce immune responses via intranasal immunization without adjuvants in comparison to that induced by h-SP and f-SP.

## 2. Materials and Methods

### 2.1. Reagents and Chemicals

Todd Hewitt broth, yeast extract, trypticase soy broth (TSB) and Bacto^TM^ agar were purchased from BD Biosciences (Franklin Lakes, NJ, USA). Blood agar plates were purchased from Hanil Komed (Seongnam, Korea). 2,2,2-Tribromoethyl alcohol, 2-methyl-2-butanol, Red Blood Cell Lysis Buffer, and hematoxylin were purchased from Sigma-Aldrich Inc. (St. Louis, MO, USA). Goat anti-mouse IgG Fc-, IgG1-, IgG2b-, and IgA-HRP were purchased from Southern-Biotech (Birmingham, AL, USA). A Mouse Dendritic Cell Enrichment Set was purchased from BD Biosciences (San Diego, CA, USA). The following antibodies were purchased from BioLegend (San Diego, CA, USA): APC-labeled anti-mouse IFN-γ, APC-labeled anti-mouse IL-4, PE-labeled anti-mouse IL-17A, PE-labeled anti-mouse CD3, APC-labeled anti-mouse CD8, PerCP-labeled anti-mouse CD11c, FITC-labeled anti-mouse I-A^b^ for MHC class II, APC-labeled anti-mouse CXCR5, PerCP-labeled anti-mouse B220, PE-labeled anti-mouse CD69, PE-labeled anti-mouse PD-1, PE-labeled anti-mouse CD80, and FITC-labeled anti-mouse CD86 antibodies. FITC-labeled anti-mouse CD4 antibody was obtained from BD Biosciences. All of the isotypes matched with each antibody were purchased from BioLegend or BD Biosciences.

### 2.2. Generation of Non-Encapsulated S. pneumoniae

The non-encapsulated form of the *S. pneumoniae* TIGR4 strain (Serotype 4) was generated as previously described [25]. Briefly, an upstream flanking region of *cps2B* was amplified using polymerase chain reaction (PCR) with the following primers: Cps4BKO-UpF: 5’-AAC TCG AGT GGA TAT CAA TTA CTA T-3’ and Cps2BKO-UpR: 5’-TTA AGC TTT CAT CTA CCC TCC ATC-3’. The amplified DNA was then digested using *Xho*I and *Hind*III. A downstream flanking region of *cps2B* was amplified by PCR using the primers: Cps4CKO-DnF: 5’-AAG AAT TCT GGT AAA AGA CTA CCG TG-3’ and Cps2CKO-DnR: 5’-TTG AAT TCT ATT TCA ACT TAC CCA AG-3’. This DNA was digested with *EcoR*I. The fragments were ligated into multiple cloning sites of pE326 [26]. Next, pKO-CPS2B and pE326, which contain upstream and downstream flanking regions of *cps2B*, were introduced into *S. pneumoniae* TIGR4 by natural transformation [27]. The deletion of C-PS was confirmed by enzyme-linked immunosorbent assay (ELISA), as previously described [28].

### 2.3. Preparation of Inactivated S. pneumoniae Whole-Cell Vaccines

Inactivated *S. pneumoniae* whole-cell vaccines were prepared as previously described [24]. Briefly, non-encapsulated *S. pneumoniae* TIGR4 was cultured in TSB at 37 °C to mid-log phase. The bacterial cells were washed twice with phosphate-buffered saline (PBS) and lyophilized. For r-SP, the lyophilized cells were irradiated with 10 kGy of gamma-irradiation using a cobalt-60 gamma-ray irradiator (point source AECL, IR-79, MDS Nordion, Ottawa, ON, Canada) at the Advanced Research Technology Institute (ARTI: Jeoneup, Korea). The cells were then resuspended in PBS. h-SP was prepared by resuspending the lyophilized cells in PBS and incubating them at 65 °C for 2 h. For f-SP, the lyophilized cells were resuspended in 0.2% formaldehyde in PBS for 2 h. The r-SP, h-SP, and f-SP were confirmed for complete inactivation by plating on agar plates of Todd Hewitt containing 0.5% yeast extract (THY) [29] at 37 °C for three days.

### 2.4. Intranasal Immunization

All of the animal experiments were conducted under the approval of the Institutional Animal Care and Use Committee of Seoul National University (SNU-171211-2). Seven-week-old male C57BL/6 mice were purchased from Orient Bio (Seongnam, Korea) and housed under specific pathogen-free conditions. After mice were subjected to anesthesia through intraperitoneal injection of avertin (2,2,2-tribromoethyl alcohol), the mice were intranasally immunized with 5 × 10^7^ CFU of r-SP, h-SP, or f-SP in 15 μL. The mice were immunized using slightly-modified methods from previous studies [24,30,31,32]. The immunization was repeated three times on days 0, 7, and 14. The control mice were immunized with an equal volume of PBS.

### 2.5. Measurement of S. pneumoniae-Specific Antibodies

Seven days after the last immunization, blood samples were obtained from the retro-orbital plexus of the mice. The sera were obtained by clotting blood samples at room temperature, then centrifuging them at 6082× *g* for 10 min. Bronchoalveolar lavage (BAL) fluid was obtained by injecting and then retrieving 500 μL of PBS into the trachea using an intravenous catheter, as described previously [33]. Nasal washes were acquired by washing the nasal cavity with 200 μL of PBS. The *S. pneumoniae* TIGR4 strain was cultured in THY and harvested at the mid-log phase. The absorbance of the pneumococcal pellet was adjusted to 0.6 of an optical density at 600 nm by dilution with PBS. A 100 μL sample of the bacterial suspension was added to 10 mL PBS. The bacterial cells were then vortexed. Next, 96-well immunoplates (SPL life science, Pocheon, Korea) were coated with 100 μL pneumococcal suspension and incubated overnight at 4 °C to allow adherence of the bacterial cells. The plates were then washed five times with PBS containing 0.05% Tween 20 (washing buffer), followed by blocking with 1% BSA in PBS containing 0.01% Tween 20 (PBS-T) for 1 h at room temperature. Next, the diluted sera were added to each well and incubated at room temperature for 1 h. The unbound antibodies were removed by washing with washing buffer. Appropriate dilutions of goat anti-mouse IgG-, IgG1-, IgG2b-, IgA-HRP were added to the wells and incubated for 30 min at room temperature. After washing the plates five times with washing buffer, 100 µl TMB substrate reagent (BD Biosciences, Franklin Lakes, NJ, USA) was added. When the colors developed, 100 µL of 2 N H_2_SO_4_ was added. Antibody titers were the reciprocal log2 of the dilution with optical density 0.2 higher than the blank measured by a microplate reader (Molecular Devices, Sunnyvale, CA, USA). Optical density at 570 nm was subtracted from that at 450 nm.

### 2.6. Analysis of the Immune Cell Population and Intracellular Cytokine Expression

In order to analyze the dendritic cell (DC) population and maturation marker expression, the mice were intranasally administered with 1 × 10^8^ CFU of r-SP, h-SP, or f-SP at a volume of 15 μL. After 24 h, the mice were sacrificed and their superficial cervical lymph nodes were harvested. Single cell suspensions were obtained from the cervical lymph nodes by passing them through a 100-μm cell strainer (BD Biosciences). The DCs were isolated from the cells by negative selection according to the manufacturer’s instructions. The expressions of CD80, CD86, and MHC II molecules were analyzed by flow cytometry. We next analyzed the immune cell population. One week after the last immunization, the cervical lymph nodes were harvested and stained with antibodies specific for CD3, CD4, CD8, CXCR5, PD-1, MHC class II, B220 or CD69, followed by analysis with flow cytometry. In order to analyze the intracellular cytokine expression, single cell suspensions (1 × 10^6^ cells/mL) were treated with PBS, r-SP, h-SP, or f-SP (1 × 10^7^ CFU/mL) for 4 days followed by stimulation with phorbol myristate acetate (PMA, 0.5 μM) and ionomycin (1 μg/mL) for 4 h in the presence of brefeldin A (1 μg/mL). The cells were fixed with PBS containing 4% paraformaldehyde for 15 min on ice and were then washed with PBS. The fixed cells were permeabilized with PBS containing 2% saponin for 15 min on ice and then stained with antibodies specific for IFN-γ, IL-4, or IL-17A. The intracellular cytokine expression was analyzed using flow cytometry.

### 2.7. Pneumococcal Infection

A single colony of *S. pneumoniae* TIGR4 was cultured in THY and harvested at its mid-log phase. *S. pneumoniae* was suspended in THY broth containing 25% glycerol, aliquoted, and stored at −80 °C until use. Before infection, the bacteria were washed three times with PBS. Ten days after the last immunization with each inactivated vaccine, the mice were intranasally infected with 50 μL of 1 × 10^4^ or 5 × 10^3^ CFU *S. pneumoniae* TIGR4. The animals’ survival was monitored daily for ten days. We also specifically analyzed the bacterial load in the lungs from mice. Two days after infection, as described above, the lungs were homogenized, serially diluted, and plated on blood agar plate to measure the bacterial load.

### 2.8. Statistical Analysis

The data are represented as mean values ± standard errors of the mean (SEM). Statistical significance was analyzed using nonparametric Kruskal–Wallis and Dunn’s post hoc (*n* < 5), one-way ANOVA and Tukey post hoc (*n* ≥ 5), or Student’s t-test. Asterisks (*) indicate statistically significant differences (*p* < 0.05) between the groups.

## 3. Results

### 3.1. Immunization with r-SP Induces Higher Protective Immunity Against Pneumococcal Systemic Infection Than Does h-SP or f-SP

We evaluated and compared the protective immunity of r-SP against its encapsulated parent strain (TIGR4) infection with those of h-SP and f-SP. Mice were intranasally immunized with r-SP, h-SP, or f-SP for three times on days 0, 7, and 14. There were no serious adverse effects of any of the vaccinations, such as mortality or weight loss (Figure 1A). Mice were then challenged with 1 × 10^4^ CFU (Figure 1B) or 5 × 10^3^ CFU (Figure 1C) of *S. pneumoniae* TIGR4 ten days after the last vaccination. Although the vaccination of r-SP increased mice survival compared to PBS group, there were no significant differences among the r-SP, f-SP, and h-SP vaccinated groups (Figure 1B,C). Furthermore, we measured the clearance of *S. pneumoniae* from the lungs by challenging the mice with 1 × 10^3^ CFU of TIGR4. Although the group that was immunized with r-SP had a significant reduction in the bacterial burden in the lungs when compared with the PBS group, there were no significant differences among the vaccinated groups (Figure 1D). These results suggest that r-SP has modest protective effect against lethal doses of pneumococcus compared to h-SP or f-SP.

### 3.2. Immunization with r-SP Enhances S. pneumoniae-Specific Antibodies in the Nasal Cavity and Lung

In order to determine whether the higher protective immunity offered by r-SP was due to the induction of the mucosal immune response, we evaluated the amount of pneumococcal-specific antibodies in the nasal cavity and lungs. The nasal wash and BAL fluid were collected from mice immunized with r-SP, h-SP, and f-SP. Significantly higher levels of IgA were observed in the nasal wash of the r-SP and h-SP vaccinated groups than in the PBS and f-SP vaccinated groups (Figure 2A). Although there was no significant difference, the mean IgA titers in the r-SP-immunized mice were slightly higher than that in the h-SP mice (*p* = 0.2217; Figure 2A). However, there were no significant levels of IgG against *S. pneumoniae* in the nasal wash from the vaccinated groups when compared to PBS group (Figure 2B). There were no significant differences in IgA titers of the BAL fluid among the groups (Figure 2C). In contrast, significantly higher IgG titer of the BAL fluid was observed in r-SP vaccinated group than in the h-SP and f-SP vaccinated groups (Figure 2D). Meanwhile, f-SP did not induce the production of pneumococcal-specific IgA or IgG in both nasal wash and BAL fluid indicating the inadequacy of formalin inactivation for developing an intranasal pneumococcal whole-cell vaccine. These findings suggest that r-SP is superior to h-SP and f-SP in promoting antibody responses in mucosal site.

### 3.3. Immunization with r-SP Increases S. pneumoniae-Specific Serum IgG and IgA

We next investigated whether the relatively higher protective response of r-SP immunization was due to the induction of serum Ig by intranasal immunization. Mice sera were collected at 7 days after the last vaccination with r-SP, h-SP, or f-SP, and pneumococcal-specific IgA or IgG titers were compared. The production of IgA was augmented in all of the vaccinated groups with no significant difference between the r-SP and h-SP vaccinated groups (Figure 3A). Interestingly, pneumococcal-specific IgG titers were potently increased in the r-SP vaccinated group compared to those in h-SP and f-SP vaccinated groups (Figure 3B). To determine the level of IgG1 and IgG2 which are respectively the surrogate marker of Th2 and Th1 response, the antibody titer induced by r-SP was compared to that by other vaccines. The production of pneumococcal-specific IgG1 was potently enhanced only by r-SP vaccination, and not by h-SP or f-SP vaccination (Figure 3C). Furthermore, IgG2b titers were also significantly increased by r-SP vaccination compared to those by f-SP vaccination (Figure 3D). Notably, we could not observe a significant titer increase of other IgG subtypes, IgG2a and IgG3, in any of the vaccinated groups (data not shown). These results suggest that r-SP is superior to h-SP and f-SP in enhancing systemic pneumococcal-specific IgG production and its subtypes IgG1.

### 3.4. Enhancement of DC Maturation by r-SP Immunization

DCs play a crucial role in adaptive immune responses. DCs capture antigens and carry them to lymph nodes to facilitate antigen-specific adaptive immune responses [34]. To further investigate the underlying mechanism for the differential immune responses to the vaccines, we measured the maturation of DCs *ex vivo*. DCs were isolated from the cervical lymph nodes of mice administered with r-SP, h-SP, or f-SP vaccines. The population of DCs was not affected by the vaccinations (Figure 4A). Although the significant differences in the expression of MHC II, CD80 and CD86 were not observed among the vaccinated groups, only r-SP remarkably induced CD80 and CD86 expression of DCs (Figure 4B–D). These results suggest that r-SP is efficient to stimulate DCs, which may contribute to the superior immunogenicity of r-SP.

### 3.5. Increases in the Populations of Follicular Helper T (T_fh_) Cells and B Cells in the Cervical Lymph Node After r-SP Immunization

We next investigated the T cell responses in the cervical lymph nodes of mice vaccinated with r-SP, h-SP, or f-SP. Seven days after the last immunization, single cells were harvested from the cervical lymph nodes and analyzed for T cell immunity. The CD4^+^ T cell populations in cervical lymph nodes were not significantly altered among the vaccinated groups, but the CD8^+^ T cell populations were significantly altered between r-SP and f-SP vaccinated group. Interestingly, the great enhancement of CD3^+^ CD4^+^ CXCR5^+^ PD-1^+^ cells (T_fh_ cell populations) of cervical lymph nodes was observed only in r-SP vaccinated group. In addition, consistent with the antibody response results, CD69^+^ MHC II^+^ B220 ^+^ cells (activated B cell populations) were also significantly enhanced only by r-SP vaccination (Figure 5A). In order to characterize the cytokine-expressing immune cells in the cervical lymph nodes, single cells harvested from cervical lymph nodes in immunized mice were stimulated with r-SP, h-SP, or f-SP for 4 days. After stimulation, each group were restimulated with PMA and ionomycin for 4 h in the presence of brefeldin A. The cells were then analyzed for the frequencies of IFN-γ-, IL-4-, and IL-17A-producing cells. Restimulation of r-SP significantly enhanced the IL-4-producing cell populations in the cervical lymph nodes immunized with r-SP, but not IFN-γ- or IL-17A-producing cell populations. Of note, the IL-17A-producing cell populations were increased only in the h-SP group after the restimulation. The immunized mice stimulated with r-SP and h-SP had slightly higher IFN-γ-producing cells, but not statistically different from PBS groups (Figure 5B). These results suggest that r-SP has superior characteristics to enhance T_fh_ cells, activated B cells, and IL-4-producing cell populations.

## 4. Discussion

*Streptococcus pneumoniae* is a major cause of severe respiratory and systemic disease in humans. It is an increasing cause of mortality and morbidity in the young and elderly. Despite the currently-available subunit vaccines for pneumococcal diseases, inactivated whole-cell vaccines are still attractive vaccine candidates. The ideal pneumococcal vaccine should be cost-effective and offer broad serotype-covering mucosal protective immunity. In this study that compares the immunogenicity of r-SP, h-SP, and f-SP, we found that the formalin inactivation method was inadequate for a pneumococcal whole-cell vaccine. The heat inactivation method may be able to elicit an effective mucosal and humoral immune responses; however, its protective immune response is limited. In contrast, the radiation-inactivation method seems to be the most suitable method to induce protective immune responses.

Previous studies have also described the effective protection of gamma-irradiated bacterial vaccines. For example, mice immunized with gamma-irradiated *Listeria monocytogenes* were better protected against *L. monocytogenes* infection compared to those immunized with heat-inactivated *L. monocytogenes* [19]. Mice immunized with irradiated *Brucella melitensis* demonstrated a higher survival rate against a lethal challenge than did those immunized with heat-inactivated one *-* [21]. Similarly, rabbits immunized with irradiated *Vibrio cholerae* showed a higher survival rate after *V. cholerae* infection than did rabbits immunized with heat- or formalin-inactivated *V. cholerae* [35]. Furthermore, mice immunized with gamma-irradiated *Staphylococcus aureus* were efficiently protected against a lethal challenge of *S. aureus* [36]. However, the immunological mechanism behind this phenomenon has not been clearly elucidated. In this study, we found that r-SP produced a higher pneumococcal-specific IgG than did other pneumococcal vaccines. The IgG response may have resulted from effective CD4^+^ T cell responses. We hypothesize that one reason for this phenomenon is that heat or formalin treatment causes modifications and/or damage to key epitopes of the pathogen, which hamper the immune response [37]. While these changes in antigenic structure can induce inaccurate and reduced protective immunity, gamma-irradiation has an advantage in that it preserves the immunogenic molecules [22].

In this study, we found that r-SP induced higher protective immune response than other vaccines. This protective immunity is supported by the presented humoral immune response results. Although we are unable to rule out the possibility of IgG leakage, our data showed that r-SP produced higher *S. pneumoniae*-specific IgA than f-SP in nasal wash and a higher *S. pneumoniae*-specific lgG than h-SP and f-SP in BAL fluid. In sera, r-SP produced higher *S. pneumoniae*-specific lgA and lgG2b than f-SP and higher *S. pneumoniae*-specific lgG, especially lgG1, than other vaccine groups. Furthermore, our previous study showed that the sera from mice immunized by r-SP have highly functional antibodies [24]. Therefore, r-SP induced higher protective immune response than other vaccines. However, in this study, although h-SP and f-SP produced less *S. pneumoniae*-specific antibody than r-SP, they also induced protective immunity against live *S. pneumoniae* in challenge model. It may be because h-SP and f-SP induced *S. pneumoniae*-specific lgA or lgG2b production *in vivo*, and h-SP or f-SP-primed BMDCs also induced cytokine production or T-cell activation *in vitro* [25]. Given that r-SP is safer to produce and induces higher protective immunity compared to other inactivated vaccines, r-SP potentially serves as a promising intranasal vaccine.

We found that r-SP and h-SP produced similar titer of *S. pneumoniae*-specific lgA antibody in nasal wash and sera, while r-SP produced higher pneumococcal-specific IgG than did other pneumococcal vaccines in BAL fluid and sera. Especially, among lgG subtypes, *S. pneumoniae*-specific IgG2b was similarly produced by r-SP and h-SP, while IgG1 was only produced by immunization with r-SP. These results agree with our previous study on immunization with r-SP, h-SP, or f-SP adjuvanted with CT. These data suggest that the higher immune responses caused by r-SP were mainly due to the antigen itself. However, in our previous study, r-SP adjuvanted with CT produced higher pneumococcal-specific IgA in BAL fluid than did other pneumococcal vaccines [24]. Therefore, CT adjuvant might be partially involved in the higher immune responses caused by r-SP.

Interestingly, we demonstrated that there were increased populations of T_fh_ cells, activated B cells, and IL-4-producing cells in the cervical lymph node of r-SP-immunized mice. These findings suggest that the T_fh_ cells are important for protective immunity against *S. pneumoniae* because T_fh_ cells stimulate the activation and differentiation of B cells into antibody-producing cells (by expressing several molecules such as inducible T-cell co-stimulator, and cytokines IL-21 and IL-4) [38]. The increased populations of T_fh_ cells by r-SP might have led to higher antibody production. However, further studies are needed to investigate whether IL-4 was directly involved in mucosal immune responses induced by r-SP. We also showed that intranasal administration of r-SP more potently induced co-stimulatory molecules and MHC II on DCs in the cervical lymph node than did the other vaccines, suggesting that DCs play a critical role in T_fh_ cell differentiation [39,40]. However, a previous report has demonstrated that gamma-irradiated *S. pneumoniae* vaccine induces IL-17-dependent protective immunity against *S. pneumoniae* [23]. This discrepancy may be due to differences in the vaccine formulation. For instance, Babb et al. used the autolysin and pneumolysin-deficient non-encapsulated *S. pneumoniae* strain derived from the parent strain D39 (serotype 2), Rx1[PdT/∆*LytA*], for generating the vaccine. In contrast, we have used the capsule-deleted *S. pneumoniae* strain derived from the parent strain TIGR4 (serotype 4). The conditions under how the vaccines were generated were also different. The Rx1[PdT/∆*LytA*] strain was irradiated with 12 kGy of gamma-irradiation using cobalt-60 gamma-ray irradiator, and kept frozen during the gamma-irradiation process [23]. In contrast, our vaccine strain (non-encapsulated *S. pneumoniae* TIGR4) was lyophilized and then irradiated with 10 kGy of gamma-irradiation using the cobalt-60 gamma-ray irradiator. These differences may explain the various intrinsic properties of the two vaccines, which may have resulted in different vaccine mechanisms.

Furthermore, γδ T cells are important initiators of the innate immune response. In a previous study [23], Babb et al. reported that their gamma-irradiated *S. pneumoniae* vaccine promotes innate γδ T-cell-derived IL-17A responses *in vivo*. However, in this study, there was no significant difference in the percentage of IL-17 producing cells between PBS and all vaccine groups. These differences may result from several reasons. First, bacteria strains used in both studies were different from each other. In the previous study, the mice were immunized with un-encapsulated D39 (serotype 2), whereas we used the un-encapsulated TIGR4 (serotype 4) in the present study. In addition, unlike their study, we used a strain that possesses pneumolysin which potently activates CD4^+^ IL-17A^+^ T cells as shown from our previous *in vitro* study [25,41]. Furthermore, while they used the lung for their analysis, we analyzed the cervical lymph node, inherently putting a difference in our result. Another major difference is in the vaccination method. We weekly administered 15 μL three times, while they administered 30 μL twice at two-week intervals. Intranasal vaccine with an amount less than 20 μL can remain confined to the upper respiratory tract, and vaccines greater than 20 uL can reach the lung [32], which may have resulted in different immune responses. Therefore, further studies are needed on the relevance of γδ T-cell in the immune responses induced by r-SP.

Our group previously showed that r-SP induced less maturation marker expression on mouse bone marrow-derived DCs than did h-SP and f-SP [25]. However, in this study, we found that the DCs from r-SP administered mice had higher expressions of CD80 and CD86 (but not MHC class II) than did those from mice vaccinated with h-SP or f-SP. The difference in the ability of r-SP to induce the maturation marker expression level in vitro and in vivo may be due to interactions between r-SP loaded DCs and other immune cells in lymphatic organs. We have also previously reported that r-SP, but not h-SP or f-SP, efficiently induces IL-8 and IL-6 production in human bronchial epithelial cells [42]. Therefore, although the direct DC activating potential of r-SP may be weaker than that of h-SP or f-SP, r-SP may potently stimulate other cell types (such as epithelial cells) that regulate immune responses in the body. In doing so, r-SP may indirectly induce the expression of maturation markers on DCs.

In this study, we compared immune responses in mice immunized with three different types of *S. pneumoniae* whole-cell vaccines. Our novel finding here was that there were increased populations of T_fh_ cells in the r-SP-immunized mice. The T_fh_ cells might stimulate B cells in the cervical lymph nodes. The increased populations of T_fh_ cells by r-SP might have led to higher production of T cell-dependent antibody responses, such as IgG. 

In summary, gamma-irradiation is suggested to be a convenient method of producing an effective pneumococcal whole-cell vaccine compared to that with conventional heat- or formalin-inactivation methods. The gamma-irradiated whole cell vaccine studied here elicited protective immunity against infections by *S. pneumoniae*.

## Figures and Tables

**Figure 1 vaccines-09-00405-f001:**
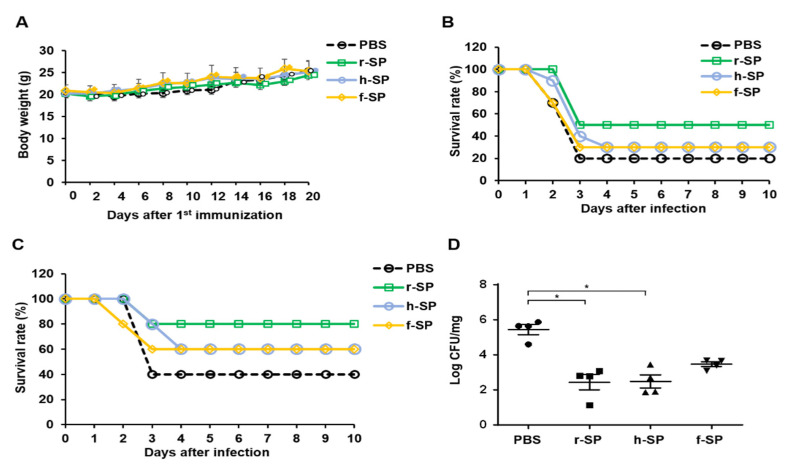
r-SP is more effective than h-SP and f-SP at intranasal immunization for protection against *S. pneumoniae* infection. C57BL/6 mice were intranasally immunized with 5 × 10^7^ CFU of r-SP, h-SP, or f-SP three times at one-week intervals. (**A**) All mice (*n* = 11 per group) were weighed every two days after vaccine administration. Weight loss was measured. (**B**,**C**) The mice were then intranasally challenged with (**B**) 1 × 10^4^ CFU (*n* = 10 per group) or (**C**) 5 × 10^3^ CFU (*n* = 10 per group) of *S. pneumoniae* TIGR4 ten days after the last immunization. Survival was monitored daily for ten days. Data were pooled from two independent experiments with ten mice or from five mice per group, respectively, and represent the survival rate of the challenged mice. (**D**) Immunized mice (*n* = 4 per group) were infected with 1 × 10^3^ CFU of *S. pneumoniae* TIGR4. Lungs were homogenized two days after the challenge to measure the bacterial load. Data are represented as mean values ± SEMs. Statistical significance was analyzed using the non-parametric Kruskal–Wallis and Dunn’s post hoc. An asterisk indicates *p* < 0.05 compared with each group.

**Figure 2 vaccines-09-00405-f002:**
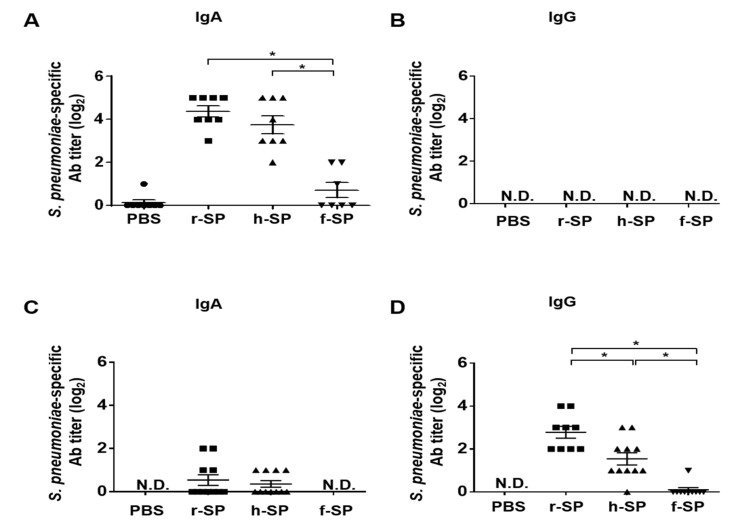
Immunization with r-SP effectively enhances *S. pneumoniae*-specific antibodies in the nasal cavity and the lung. C57BL/6 mice were intranasally immunized with 5 × 10^7^ CFU of r-SP, h-SP, or f-SP three times at one-week intervals. At seven days after the last immunization, the nasal wash and BAL fluid were collected from the immunized mice. *S. pneumoniae*-specific (**A**) IgA (*n* = 7–8 per group) and (**B**) IgG (*n* = 3–6 per group) titers in the nasal wash and *S. pneumoniae*-specific (**C**) IgA (*n* = 8–11 per group) and (**D**) IgG (*n* = 9–12 per group) titers in the BAL fluid were examined using ELISA. The data are expressed as geometric mean antibody titers ± SEMs. Differences in the antibody titer between each group were analyzed using one-way ANOVA and Tukey post hoc. An asterisk indicates *p* < 0.05 compared with each group. N.D. denotes not detected.

**Figure 3 vaccines-09-00405-f003:**
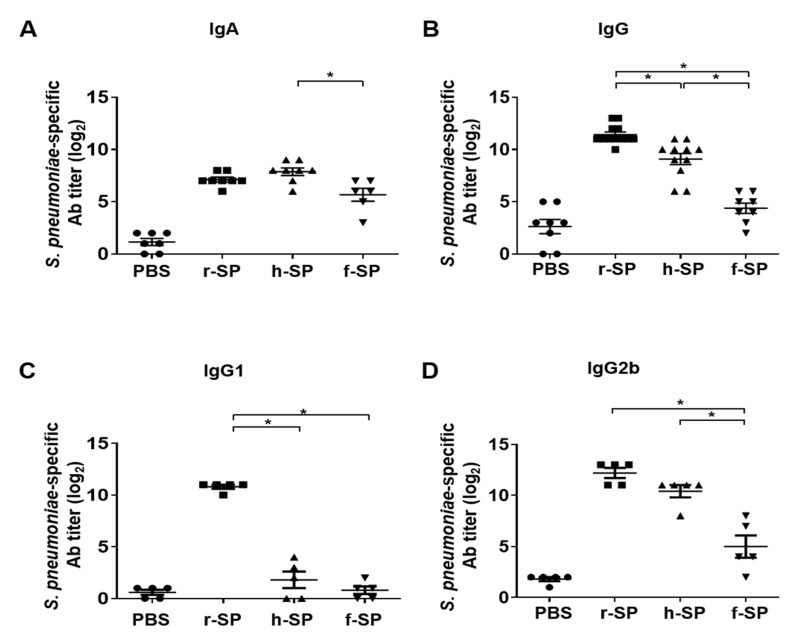
Immunization with r-SP effectively enhances *S. pneumoniae*-specific antibodies in the serum. C57BL/6 mice were intranasally immunized with 5 × 10^7^ CFU of r-SP, h-SP, or f-SP three times at one-week intervals. The sera were collected from mice at seven days after the last immunization. *S. pneumoniae*-specific (**A**) IgA (*n* = 6–8 per group), (**B**) IgG (*n* = 8–12 per group), (**C**) IgG1 (*n* = 5 per group), and (**D**) IgG2b (*n* = 5 per group) were examined using ELISA. The data are expressed as geometric mean antibody titers ± SEMs. Differences in antibody titers between each group were analyzed using one-way ANOVA and Tukey post hoc. An asterisk indicates *p* < 0.05 compared with each group.

**Figure 4 vaccines-09-00405-f004:**
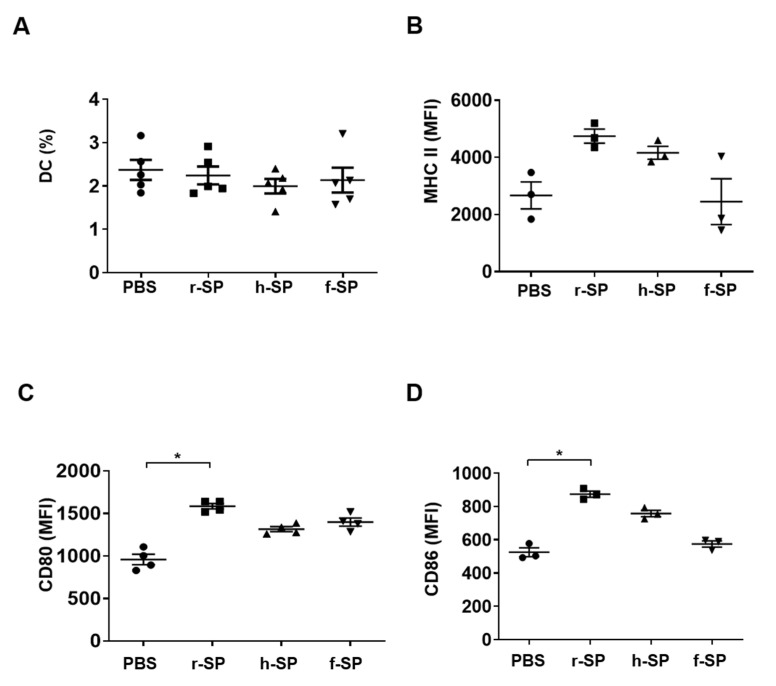
Immunization with r-SP enhances the maturation of DCs in the cervical lymph node. C57BL/6 mice were intranasally administered with 1 × 10^8^ CFU of r-SP, h-SP, or f-SP. Then, 24 h after the administration, the cervical lymph nodes were harvested and checked for size (*n* = 5 per group). (**A**) A single cell suspension from the cervical lymph node was stained with fluorochrome-conjugated monoclonal antibodies specific for MHC class II and CD11c to determine the percentage of DC population. The population was then analyzed using flow cytometry to obtain the mean values of MHC class II^+^ CD11c^+^ cells ± SEM. (**B**–**D**) After the DCs were isolated from the cervical lymph nodes of each vaccine-administered mouse, the expression of (**B**) MHC class II (*n* = 3 per group), (**C**) CD80 (*n* = 4 per group), and (**D**) CD86 (*n* = 3 per group) on DCs was analyzed by flow cytometry to obtain the mean fluorescence intensity (MFI) ± SEM. Differences between each group were analyzed using the one-way ANOVA and Tukey post hoc (*n* ≥ 5) or the nonparametric Kruskal–Wallis and Dunn’s post hoc (*n* < 5). Asterisks indicate *p* < 0.05 compared with each group.

**Figure 5 vaccines-09-00405-f005:**
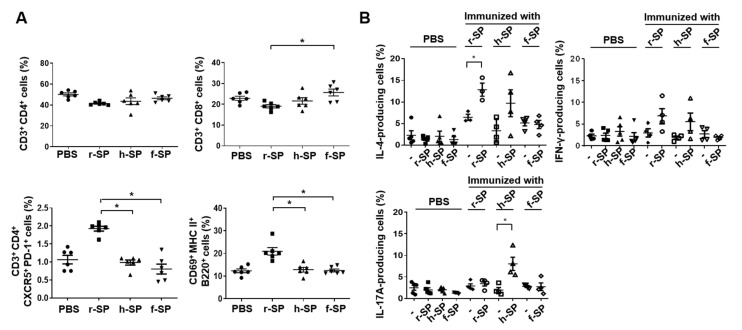
Immunization with r-SP increases T_fh_ cell and B cell populations and IL-4-producing cell frequency in the cervical lymph node. C57BL/6 mice were intranasally immunized with 5 × 10^7^ CFU of r-SP, h-SP, or f-SP three times at one-week intervals. (**A**) Single cell suspensions from the cervical lymph nodes were stained with fluorochrome-conjugated monoclonal antibodies specific for CD3 and CD4, CD3 and CD8, CXCR5 and PD-1, or MHC class II and CD69 followed by flow cytometry (*n* = 6 per group). Differences between each group were analyzed using the one-way ANOVA and Tukey post hoc. Asterisks indicate *p* < 0.05 compared with each group. (**B**) Single cells (1 × 10^6^ cells/mL) were stimulated with or without r-SP, h-SP, or f-SP (1 × 10^7^ CFU/mL) for 4 days (*n* = 3–5 per group) and then, the unstimulated- or inactivated whole cell-stimulated cells were restimulated with PMA (0.5 μM) and ionomycin (1 μg/mL) for 4 h in the presence of brefeldin A (1 μg/mL). After harvesting the cells, the frequencies of IL-4-, IFN-γ-, and IL-17A-producing cells were analyzed by flow cytometry. Scatter plot represents the mean values of the percentage of cell populations ± SEM. Differences between each group were analyzed using Student’s *t*-test. An asterisk indicates *p* < 0.05 compared with each group.

## Data Availability

The data that support the findings of this study are available on request from the corresponding author.

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
