# Peer review of "Immune Responses to Irradiated Pneumococcal Whole Cell Vaccine"

_vaccines, 2021, doi:10.3390/vaccines9040405_

Round 1

Reviewer 1 Report

Summary

Streptococcus pneumoniae continues to be a global health burden for the elderly, children and immunocompromised people leading to both respiratory and systemic diseases. Currently available vaccines either don’t provide long term protection or are too costly for global use. The intranasal administration of the non-encapsulated, gamma-irradiated r-SP vaccines have shown promise inducing both local and systemic immune responses. However, the immunological mechanism of the gamma-irradiated vaccine compared to the heat-inactivated or the formalin-inacativated versions are unknown. This manuscript demonstrates the difference between the 3 formulations suggesting why the gamma-irradiated vaccine is superior in effectiveness compared to the others.

Abstract

Line 22: The abstract indicates that there were two vaccinations events but in the methods they are vaccinated three times over a two week period.

Line 23: Cellular immunity was indicated as being investigated however there is little data to suggest that cellular immunity was the focus of the work in this manuscript.

Introduction

Line 87-88: if the authors were curious to determine whether the immune responses were from r-SP or r-SP or the adjuvanted formula then it should have been used as a positive control.

Materials and methods:

Line 137: what was the type of anaesthesia used?

Line 138: why was the volume changed to 15 uL from the authours’ previous publication of 16 uL? Was the volume administered in one nare or evenly in both nares? The volume used depending on the method of administration to the nose would result in either facial or upper respiratory mucosal results. Was the intranasal vaccination protocol optimized? Would there have been a stronger antibody response 2 weeks after the final vaccination?

Line 160-162: how were the titres determined using the OD?

Results:

Line 222-226: Figure 1D, the data are represented as an n=4, how were the normality tests performed with a low number? The statistics using only pair comparisons to the control and parametrically need to be explained. The statistics should be done with a nonparametric ANOVA and have the post hoc analyses compare all groups.

Line 246-24: Figure 2A/2D, the statistics as they are would need to be justified as they should be performed as an ANOVA (after normality tests were determined) with an appropriate post-hoc evaluation.

Line 270-276: Figure 3A-D, please see previous comments regarding statistics.

Line 289-301: Figure 4A-D. Please verify the y-axis for 4A as it indicates DC% but is referring to size in the figure legend. Please see the comments regarding statistics and multiwise comparisons.

Line 323-334: Figure 5A, B. Would strongly suggest that Figure 5A be changed to clearly indicate the specific populations being determined by flow cytometry. Which markers were used for activated B cells? Please see above comments for statistics for Figure 5A only, 5B is fine.

The hallmark of mucosal immunity is antigen-specific IgA in either the BAL or the nasal wash. Why was there as lack of IgA in the lung? How can be it be determined that the IgG present in the lung was mucosal and not the result of IgG leaking across the mucosa? How are these results different from the adjuvanted formula that was previously published?

Why were the other IgG isotypes not investigated when assessing humoral immune responses?

Was flow cytometry done to validate the purity of the dendritic cell population after the negative isolation?

Which cervical lymph nodes were collected? Superficial, deep or both? If expecting lung antibody responses why were the mediastinal lymph nodes not assessed?

Why were the cytokines IL-4, IFN-g and IL-17A selected for the restimulation assays?

Discussion

The discussion and in fact the manuscript would be stronger if the vaccine without the adjuvat was discussed in relation to the adjuvanted with CT version as was first mentioned in the introduction.

Line 370-371: if associating IL-4 with the responses seen in the mucosa it would be important to demonstrate that there isn’t a significant amount of antigen-specific IgE or add the other cytokine of importance for IgA, TGF-beta.

Line 374-376: references are needed for this statement.

Gamma-delta T cells are considered to be initiators in the innate immune response and the IL-17 mentioned in reference 24 is innate. How does this conflict with your results when there is not gamma-delta specific T cell information?

Again, there are very little results to suggest the cellular immunity of the formulations.

There is a lack of discussion regarding the results determined for the humoral immune response and the protection seen between the vaccine groups after challenge. There is no significant difference between groups for protective immunity but consistently there are significant differences between the r-SP, h-SP and f-SP.

Reviewer 2 Report

The manuscript presented by Ko et al. is a nice continuation on their previous work with irradiated S. pnuemoniae. While the first publication compared the efficacy of irradiated S. pneumoniae with other inactivated S. pneumoniae administered with adjuvant, the present study was performed using the inactivated bacteria only, without adjuvant. The findings confirm their previous report on the superiority of irradiation-inactivated S. pneumoniae and provides further details regarding the underlying mechanism.

One minor issue that I noticed is the inappropriate use of the Student's t test for statistical analysis. Since more than two groups are compared in the analysis, the authors should apply ANOVA testing.

Round 2

Reviewer 1 Report

Thank you to the authors for taking our suggestions into consideration and modifying your manuscript as previously outlined.

There are a few small changes that we suggest before publication:

  1. Line 270-272: should clarify that they were looking for IgG1 and IgG2 for Th2 and Th1 respectively.
  2. Figure 4D, suggest the scale should be the same as the others in Figure 4, starting at 0 on the y-axis.
  3. In Figure 5 for the Tfh data, of the error bars appear smaller, was the figure +/-SD previously and is now +/-SEM?
  4. The review suggests that the new sections that have been added in require some English language editing. For example, line 391 would suggest changing ‘although h-sp and f-sp less produced S. pneumoniae…’ to although h-sp and f-sp produced less S. pneumoniae…’
